# A naturally protective epitope of limited variability as an influenza vaccine target

Craig P. Thompson [1,2], José Lourenço [1], Adam A. Walters[2], Uri Obolski[1], Matthew Edmans[1,2], Duncan S. Palmer [1], Kreepa Kooblall[3], George W. Carnell[4], Daniel O'Connor [5], Thomas A. Bowden[6], Oliver G. Pybus[1], Andrew J. Pollard[5], Nigel J. Temperton [4], Teresa Lambe[2], Sarah C. Gilbert [2] & Sunetra Gupta[1]

Current antigenic targets for influenza vaccine development are either highly immunogenic epitopes of high variability or conserved epitopes of low immunogenicity. This requires continuous update of the variable epitopes in the vaccine formulation or boosting of immunity to invariant epitopes of low natural efficacy. Here we identify a highly immunogenic epitope of limited variability in the head domain of the H1 haemagglutinin protein. We show that a cohort of young children exhibit natural immunity to a set of historical influenza strains which they could not have previously encountered and that this is partially mediated through the epitope. Furthermore, vaccinating mice with these epitope conformations can induce immunity to human H1N1 influenza strains that have circulated since 1918. The identification of epitopes of limited variability offers a mechanism by which a universal influenza vaccine can be created; these vaccines would also have the potential to protect against newly emerging influenza strains.

[1] Department of Zoology, University of Oxford, Oxford OX1 3PS, UK. [2] The Jenner Institute Laboratories, University of Oxford, Oxford OX3 7DQ, UK. [3] Oxford Centre for Diabetes, Endocrinology and Metabolism, Churchill Hospital, University of Oxford, Oxford OX3 7LE, UK. [4] Medway School of Pharmacy, University of Kent, Chatham ME4 4BF, UK. [5] Oxford Vaccine Group, Department of Paediatrics, University of Oxford, and the NIHR Oxford Biomedical Research Centre, Oxford OX3 7LE, UK. [6] Division of Structural Biology, Wellcome Centre for Human Genetics, University of Oxford, Oxford OX3 7BN, UK. Correspondence and requests for materials should be addressed to C.P.T. (email: craig.thompson@zoo.ox.ac.uk) or to S.G. (email: sunetra.gupta@zoo.ox.ac.uk)

Seasonal influenza is estimated to cause between 1 and 4 million cases of severe illness and 200,000 to 500,000 deaths per year[1]. The best way to protect against influenza infection is through vaccination. Currently, a trivalent (TIV) or quadravalent influenza (QIV) vaccine is given each year, targeting the circulating H1N1 and H3N2 influenza A strains and one or two lineages of the circulating influenza B strains. However, the vaccine has to be formulated at least 6 months prior to the influenza season and so the strains that are subsequently prevalent in the actual flu season do not always match the strains used in the vaccine[2].

The antigenic evolution of influenza is known to occur through mutations in surface glycoproteins, principally haemagglutinin (HA), allowing strains to escape the pre-existing host immunity[3–5]. Epitopes within HA are commonly assumed to be either highly variable due to strong immune selection (and typically located in the head domain of HA) or conserved due to the absence of immune selection (for example, in the stalk of HA)[6]. Together, these form the backbone of the theory of antigenic drift, whereby the virus population slowly and incrementally acquires mutations in protective highly variable epitopes. However, the antigenic drift model can only explain the epidemiology and limited genetic diversity observed among influenza virus populations when very specific constraints are placed on the mode and tempo of mutation or by invoking short-term strain-transcending immunity[7,8]. An alternative model known as antigenic thrift successfully models the epidemiology and genetic diversity of influenza by assuming that immune responses against epitopes of limited variability drive the antigenic evolution of influenza[9–11]. Within this framework, new strains may be generated constantly through mutation, but most of these cannot expand in the host population due to pre-existing immune responses against epitopes of limited variability. This creates the conditions for the sequential appearance of antigenically distinct strains and provides a solution to the long-standing conundrum of why the virus population exhibits such limited antigenic and genetic diversity within an influenza epidemic. An important translational corollary of this model is that a universal influenza vaccine may be constructed by targeting such protective epitopes of limited variability.

We show that studies of sera from young children taken in 2006/7 using neutralisation assays and ELISAs reveal a periodic pattern of cross-reactivity to historical isolates consistent with the recycling of epitopes of limited variability. We identify one epitope of limited variability responsible for this pattern through a structural bioinformatics analysis. We demonstrate that mutagenesis of the epitope removes the cross-reactivity to historical strains, and vaccination of mice with the 2006 conformation of the epitope is able to reproduce the cross-reactivity pattern identified in the serology studies. We further show that vaccination of other epitope conformations induces similar but asynchronous cross-reactivity to historical strains. Finally, we demonstrate that vaccination with either the 2006 or 1977 epitope conformations is able to protect the mice from the challenge with a H1N1 influenza strain that last circulated in 1934. By establishing that the antigenic space within which influenza evolves is much smaller than previously thought, we show that there are epitopes in the major influenza antigen, HA, which if vaccinated against would allow us to avoid the requirement for yearly influenza vaccination necessitated by the current TIV and QIV vaccines.

## Results

**Periodic cross-reactivity to historical isolates.** We tested the prediction that HA epitopes of limited variability exist by performing microneutralisation assays using pseudotyped lentiviruses, displaying the H1 HA proteins from a panel of historical influenza isolates (hereafter described as pMN assays[12,13]), with sera obtained in 2006/2007 in the UK from 88 children born between March 1994 and May 2000 (Fig. 1a). This panel consisted of pseudotyped lentiviruses displaying HAs from seasonal influenza strains as well as two pandemic strains: A/California/04/2009 and A/South Carolina/1/1918. All individuals possessed neutralising antibodies to the A/Solomon Islands/3/2006 strain, and 98% of individuals possessed neutralising antibodies to A/New Caledonia/20/1999. A total of 99% of individuals also possessed neutralising antibodies to the A/USSR/90/1977 strain, while 30% of individuals also possessed neutralising antibodies to the A/WSN/1933 strain. By contrast, only 3%, 9.1% and 3.4% of individuals possessed antibodies to the A/California/4/2009, A/PR/8/1934 and A/South Carolina/1/1918 strains, respectively. ELISA analysis using the HA1 domain of the same seven strains as an antigen was consistent with the pMN data and also identified broadly cross-reactive non-neutralising antibodies that bind the HA1 region of various H1 influenza strains (Supplementary Fig 1). These results are in agreement with a number of recent studies suggesting that antibody responses showing some degree of periodic cross-reactivity to historical strains[14–22] counter to the view of antigenic drift within which antigenic distance accumulates linearly with time.

We noted that the A/Solomon Islands/3/2006, A/New Caledonia/20/1999, A/PR/8/34 and A/WSN/33 HAs all contained a deletion at position 147 (linear numbering, where Met = 1, H3 numbering: 133, WHO numbering: 130) and exhibited some degree of reactivity to the 2006/2007 sera. This position otherwise typically contains a positively charged arginine, in the case of the A/USSR/90/1977 HA, or lysine, as is the case for the A/California/4/2009 and A/Brevig Mission/1/1918 HAs. To determine whether the cross-reactivity observed between these strains could be attributed to the deletion, a lysine was inserted at position 147 (−147K) in the A/Solomon Islands/6/2006, A/PR/8/1934 and A/WSN/1933 HAs (Fig. 1b–d). A statistically significant loss of neutralisation for the −147K A/Solomon Islands/3/2006, A/PR/8/1934 and A/WSN/1933 mutant pseudotyped lentiviruses was observed ($p$-value = 0.0005, Fig. 1b, $p$-value = 0.0004, Fig. 1c and $p$-value = 0.0056, Fig. 1d; $p$-values were determined using Student's $t$ tests). In the case of the A/PR/8/1934 −147K mutant, there was a total loss of neutralisation in four samples and a reduction in two samples indicating that the bulk of cross-reactivity between the A/Solomon Islands/3/2006 and the A/PR/8/1934 strains is mediated through an epitope located in the vicinity of the deletion ($p$-value = 0.0004 using Student's $t$ test, Fig. 1c). Therefore, it seems that the absence of a positively charged lysine at position 147 mediates much of the observed cross-reactivity to historical strains induced by the 2006/2007 cohort sera in Fig. 1a.

Analysis of historical strain data shows that the deletion, the only one to occur in the H1 HA, appears periodically over the course to the antigenic evolution of H1N1 subtype of influenza occurring in 1933, 1934, 1943, 1957 and between 1995 and 2008. To ascertain whether the absence of an amino acid at position 147 would also mediate cross-reactivity with other historical strains possessing the deletion, A/WSN/1933 neutralisation-positive samples were run against the WT and −147K mutant A/Iowa/1943 and A/Denver/1957 pseudotyped lentiviruses (Fig. 1e, f). A statistically significant reduction in neutralisation was observed for the −147K A/Iowa/1943 and A/Denver/1957 HA mutants ($p$-value = 0.012, Fig. 1e, $p$-value = 0.011 using Student's $t$ test, Fig. 1f). Furthermore, three samples, which neutralised the WT A/Denver/1957 HA failed to neutralise the −147K mutant entirely. These results imply that at least part of the cross-

reactive neutralising immune response within this cohort is mediated through the recognition of an epitope that contains a deletion at position 147. Moreover, the existence of a lysine at position 147 may contribute to the overall lack of neutralisation of A/California/4/2009 and A/South Carolina/1/1918, in addition to other variation across the HA.

**Identification of an epitope of limited variability.** Several previous studies have highlighted the importance of position 147. Although not included within any of the canonical antigenic sites defined by Caton et al.[3] (being absent in the A/PR/8/1934 Mt. Sinai strain), position 147 has recently been assigned to a new

antigenic site-denoted Pa in Matsuzaki et al.[23], where it was shown to be responsible for several A/Narita/1/2009 escape mutants[3,23]. Position 147 is also important for the binding of several known broadly neutralising antibodies: for example, the 5j8 antibody requires a lysine to be present at position 147, whilst the CH65 antibody cannot bind if a lysine is present at position 147[24,25]. Furthermore, Li et al.[18] demonstrated that certain demographics, such as individuals born between 1983 and 1996, possess antibodies that bind to an epitope containing a lysine residue at position 147[18].

We next employed a structural bioinformatic approach to identify an epitope of limited variability that contained position

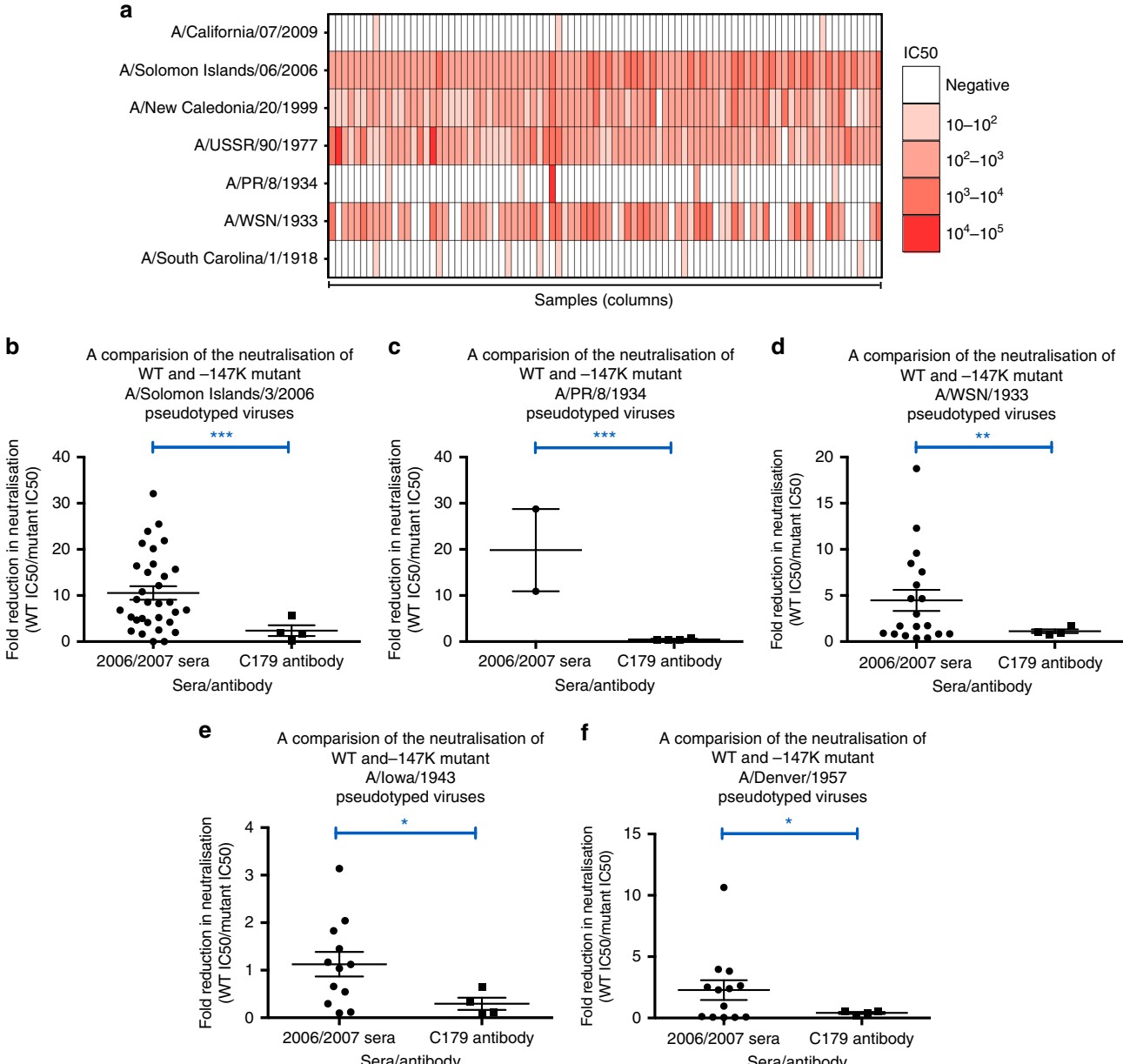

**Fig. 1** Pseudotype microneutralisation data reveals a cyclic pattern of epitope recognition. **a** Serum samples from children aged between 6 and 12 years in 2006/2007. $n = 88$ were tested for their ability to neutralise a panel of pseudotyped lentiviruses representing a range of historical isolates. **b**–**f** A lysine residue was inserted at position 147 linear numbering, where Met = 1 through site-directed mutagenesis SDM in the HAs of pseudotyped lentiviruses A/WSN/1933, A/PR/8/1934 and A/Solomon Islands/3/2006 included in panel A as well as A/Iowa/1943 and A/Denver/1957. The ratio of WT IC50 to mutant IC50 was then assessed to determine if there was reduction in neutralisation. A stalk-targeting antibody, C179, was used as a control. The values provided are an average of two replicates. Student's $t$ tests were used to determine all $p$-values: ***$p$-value < 0.001, **$p$-value < 0.010 and *$p$-value < 0.050. Error bars are mean ± s.e.m.

147. In silico analysis was used to determine how the accessibility and binding site area contributed to the variability of hypothetical antibody binding sites (ABSs) (Fig. 2a) on the surface of the A/Brevig Mission/1/1918, A/PR/8/1934, A/California/04/2009 and A/Washington/5/2011 H1 HA crystal structures[26–30]. The ABS of lowest variability containing position 147 was consistently represented by the site shown in Fig. 2b[31] and could be shown to locate to an exposed loop in the head domain of the H1 HA, not covered by N-linked glycosylation (Fig. 2b, c), and encompassing additional residues in the Ca$_2$ antigenic site.

Analysis of this site (hereafter called OREO) suggested that various conformational epitope variants could be defined on the basis of variation and structural proximity of positions 147, 158 and 159. Combining these analyses with the site-directed mutagenesis SDM results, we arrived at a maximum of five epitope conformations of the epitope (Fig. 3 and Supplementary Fig 3), which arise and disappear in a cyclical manner during the known evolutionary history of the pre-pandemic and post-pandemic H1N1 lineages (Fig. 2d). This analysis demonstrates that there are numerous potential sites of limited variability in the head domain of the H1 HA represented by the local minima in Fig. 2a, in addition to a range of highly variable sites; the antigenic trajectory of the latter has been tracked in detail by several previous studies[32,33].

**Vaccination induces cross-reactivity to historical isolates**. We next substituted the five proposed conformations of OREO (Fig. 3 and Supplementary Table 1) into H5, H6 and H11 HAs, which have not circulated in the human population, to maintain the conformations of the epitope and allow the immune response to be focused on the epitope via a prime–boost–boost vaccination regimen (Fig. 4a). Consequently, mice were vaccinated using a DNA–DNA-pseudotyped lentivirus regimen alternating between different HA scaffolds (Fig. 4a). Analysis of sera obtained from the final bleed at 21 weeks prior to the influenza challenge demonstrated that vaccinating with these epitopes produces antibodies that are cross-reactive to a number of historical strains. Notably, the 2006-like epitope conformation (red) produces cross-reactive antibodies that mirror the neutralisation profile of sera taken in 2006/2007 from young children aged 6–12 years (Fig. 1a and Supplementary Fig 1): both datasets show neutralisation of pseudotyped lentiviruses displaying HAs from A/Solomon Islands/3/2006, A/USSR/90/1977, A/PR/8/1934 and A/WSN/1933 via the OREO epitope, but not from A/California/4/2009 or A/South Carolina/1/1918 (Fig. 4b–g). Intriguingly, the 1977-like conformation (green) containing an arginine at position 147 also displays similar cross-reactivity to that of the 2006-like epitope (red), containing a deletion at position 147. Furthermore, the 2009-like (blue) and 1991-like (orange) conformations showed periodic cross-reactivity to historical strains demonstrating the chronological reoccurrence of epitopes of limited variability (Fig. 4b–g).

**Vaccination protects against heterologous challenge**. To test whether antibodies directed against these epitopes conferred protective immunity, the 2006-like (red) and 1977-like (green) epitope-vaccinated groups were challenged with a strain collected in 1934 (A/PR/8/1934) (Fig. 5a, c). The 2009-like (blue), 1995-like (orange) and 1940-like (pink) groups were challenged with a 2009 pandemic strain (A/California/04/2009; Fig. 5b, d). In each challenge experiment, an unvaccinated group ($n = 6$) was included as well as a group vaccinated via the DNA–DNA-pseudotyped lentivirus regimen with the H6, H5 and H11 HAs without the substituted epitope conformations ($n = 6$). Vaccination with the 2006-like (red) and 1977-like (green) epitope conformations

conferred immunity to challenge with the A/PR/8/1934 virus (Fig. 5a, c). As expected, vaccination with the 2009-like epitope also conferred immunity to challenge with A/California/04/2009 strain, which last circulated in 2009 (Fig. 5b, d). These results demonstrate that epitopes that circulated in the A/Solomon Islands/8/2006 and A/USSR/90/1977 strains, which last circulated in 2006 and 1977, respectively, were able to produce antibodies that confer protection against challenge with the A/PR/8/1934 strain, which last circulated in 1934.

**Discussion**

Our results demonstrate the existence of a highly immunogenic epitope of limited variability in the head domain of the H1 HA, which has been theorised by mathematical modelling studies to drive the antigenic evolution of influenza[9,10]. Sera from children aged 6–12 years taken in 2006/7 were shown to cross-react with a panel of historical isolates, the majority of which they will not have experienced (Fig. 1a). This cross-reactivity was removed by mutagenesis of an epitope of limited variability identified through a structural bioinformatic analysis (Fig.1b–f and Fig. 2). We were further able to reproduce the cross-reactivity exhibited in the serology studies in a mouse model, and demonstrated that vaccination with the epitope conformations circulating in 2006 or 1977 induced protective immunity to challenge with a strain that last circulated in 1934 (Fig. 4). Vaccination with other conformations of the epitope produced complementary but asynchronous cross-reactivity to historical strains (Figs. 4 and 5). Furthermore, between 1918 and the present day, the 2006-like 147-deleted conformation of the epitope has occurred 5 times. In two instances, when circulating strains contained the 147-deleted conformation of the epitope, lineage replacement of the H1N1 strain occurred (in 1957 and 2008). This suggests that the possession of a conformation of the epitope in which 147 is present conferred a very significant selective advantage once population immunity has built up against the 147-deleted conformation.

Another site of limited variability identified by our analysis in the head domain of the H1 HA appears to be centred on position 180 (linear numbering, position 166: H3 numbering, position 163: WHO numbering). Linderman et al[19]. and Huang et al[20]. have identified and purified antibodies that bind to a site including position 180 and neutralise A/California/04/2009, A/USSR/07/1977 and A/Brevig Mission/1/1918, but not A/Solomon Islands/30/2006, A/New Caledonia//1999 and A/PR/8/1934, displaying cyclical cross-reactivity in a similar but asynchronous manner to OREO[19,20,22]. However, as this site is periodically covered by glycosylation, the OREO epitope is likely to be a better vaccine target.

Currently available influenza vaccines are believed to target epitopes of very high variability on the haemagglutinin and neuraminidase surface glycoproteins. This requires them to be continuously updated, with the only alternative being seen as the artificial boosting of immunity to conserved epitopes of low immunogenicity. By identifying such epitopes, we have established an alternate method of producing improved influenza vaccines: targeting highly immunogenic epitopes of limited variability as opposed to targeting highly immunogenic epitopes of high variability or conserved epitopes of low immunogenicity. Through vaccination against the various conformations of the epitope of limited variability identified in this study, it is possible to induce immunity to all previous and future H1N1 strains.

The OREO site has been under selective pressure over a period of 80 years, providing us the opportunity to observe and document its variation: the site has historically cycled between four conformations when position 147 is present and one

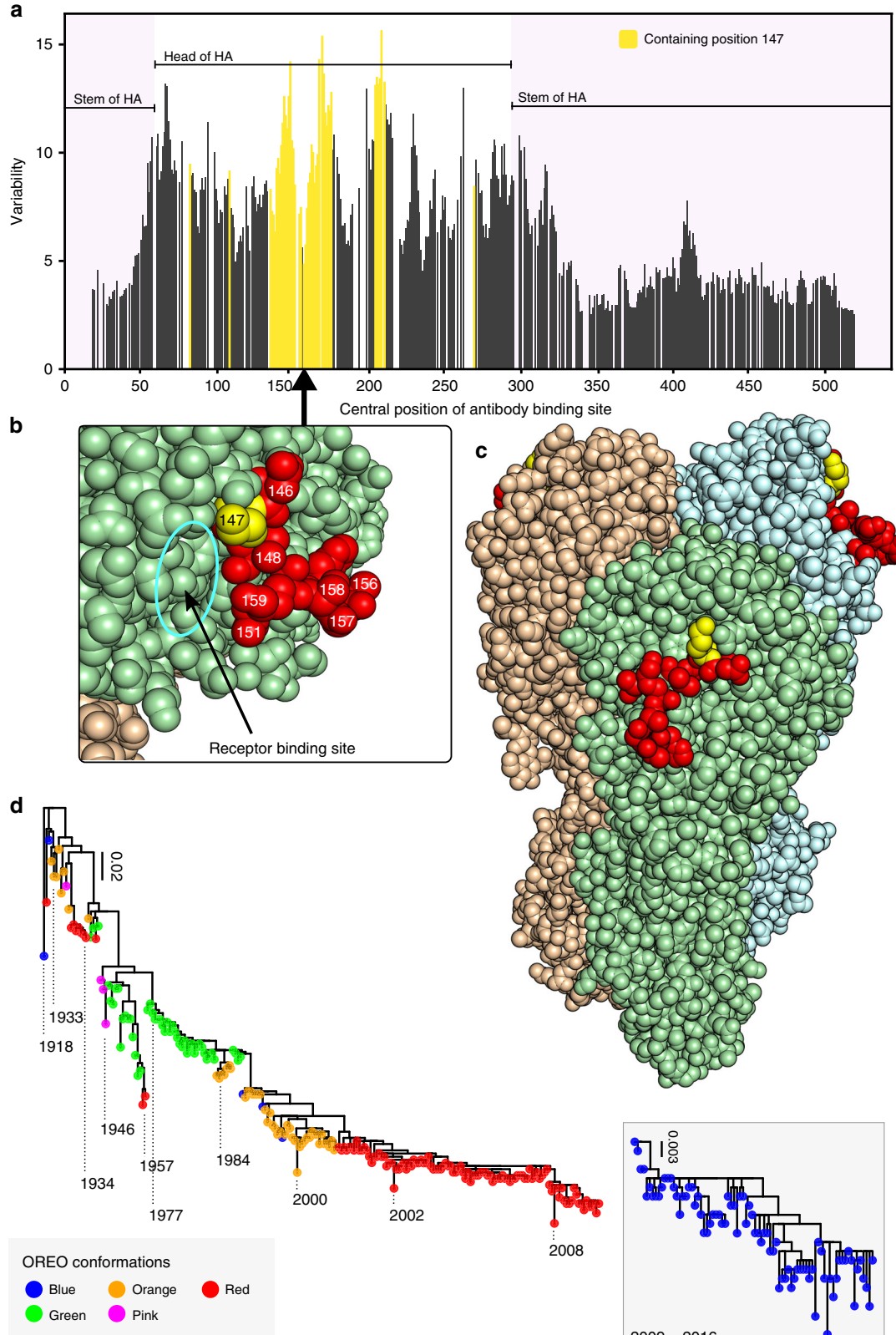

**Fig. 2** Identification of a site of limited variability in the head domain of the H1 HA. **a** Variability of antibody bindings sites (ABSs) on the crystal structure of A/California/04/2009 HA; those containing position 147 are shown in yellow. **b**, **c** Location of ABS of lowest variability containing position 147 with position 147 shown in yellow and the rest of the site coloured in red. **d** Phylogenetic trees of pre-pandemic and post-pandemic highlighted rectangle H1N1 with tips coloured according to the conformation of the epitope of limited variability (hereafter called OREO). Please note the re-introduction of H1N1 influenza in 1977 involved a strain which previously circulated in 1949/50[39, 40]

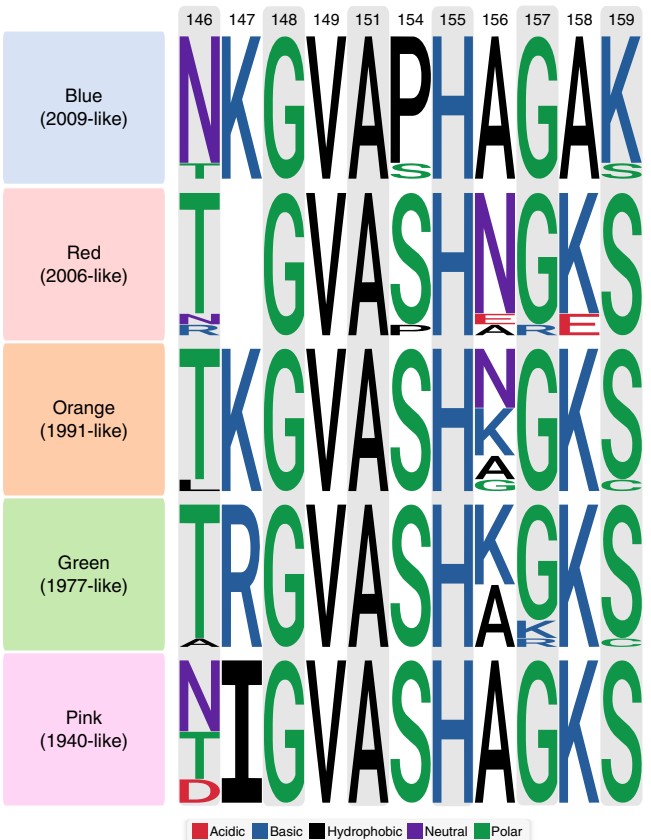

**Fig. 3** Allelic classes of the OREO epitope. Sequence Logo diagrams showing the relative frequency of different amino acids for each OREO conformation based on yearly consensus sequences. The various epitopes can be defined by the amino acids at positions 147, 158 and 159. These three positions have been used to define the conformations of OREO in the phylogenetic tree in Fig. 2d

conformation where it has undergone a deletion. Once the deletion has occurred, instead of the epitope varying further, on two occasions the circulating seasonal H1N1 strain died out—once in 1957 and again in 2009. In 2009, the seasonal influenza strain was replaced through zoonotic spillover with a pandemic H1N1 strain displaying a previously seen conformation of OREO, to which immunity in 2009 has not yet built up against[34,35]. In 1957, the seasonal influenza strain was replaced with H2N2 influenza (Fig. 2d)[36]. These observations suggest that the antibodies generated against these five conformations cover all possible variations within the OREO epitope that are found in evolutionary fit H1N1 influenza viruses.

This limited variation exhibited by the epitope is determined by the composition of the site. Position 147 is next to, and affects the polarity of, the receptor binding site, and therefore there is a limited repertoire of amino acids it can cycle between. Furthermore, the OREO site is composed of a smaller number of residues compared with other sites containing position 147, which intrinsically limits its overall variability (Supplementary Fig 4a, b).

All the conformations of the OREO epitope could be displayed in a single vaccine by itself or in concert with other epitopes of limited variability to create a single universal influenza vaccine. Alternatively, the epitopes could be displayed in individual vaccines and deployed when the circulating conformation changes (currently the OREO epitope changes roughly every 10 years, see

Supplementary Fig 3). The critical feature of both approaches described above is that the OREO epitope does not drift to the same level as the currently targeted highly variable epitopes. Hence, both of these approaches could provide longer-lasting vaccines in comparison to the trivalent and quadrivalent vaccines. Moreover, the longer cycling period of OREO might help decrease and also avoid mistakes in the formation of the vaccines, which sometimes occur due to formulation decisions having to be made at least 6 months prior to the influenza season[2].

The evolutionary framework on which these studies are based[9,37] applies generally to other subtypes of influenza A such as H3N2[37] and also to influenza B, suggesting that epitopes of limited variability can also be identified in these viruses. Indeed, Zinder et al.[37] have shown that the phylodynamics of H3N2 influenza is easily reproduced using the antigenic thrift framework. Consequently, the same strategy could be used to produce vaccines for other subtypes of human influenza, as well as swine and avian influenza viruses and potentially other viruses.

## Methods

**Serum samples.** A total of 88 serum samples from young children aged 6 to 12 years were collected in Oxford in 2006/2007. The legal guardians of all donors gave written informed consent for research use of blood samples with ethical approval by a local research ethics committee, South Central– Hampshire B Research Ethics Committee (ref: 16/SC/0141). Further permission for the use of the samples in this specific study was also granted by the Oxford Vaccine Centre Biobank.

**Enzyme-linked immunosorbent assay (ELISA).** Anti-HA1 antibody responses were measured using ELISAs. In brief, Nunc-Immuno 96-well plates (Thermo Fischer Scientific, USA) were coated with 1.0 µg ml$^{-1}$ of HA1 protein (Sino Biological Ltd, China) in PBS buffer and left overnight at 4 °C. Plates were washed with 6x with PBS–Tween PBS/T, then blocked with casein in PBS for 1 hour at room temperature. Serum or plasma was diluted in casein–PBS solution at dilutions ranging from 1:50 to 1:1000 before being added to Nunc-Immuno 96-well plates in triplicate. Plates were incubated at 4 °C overnight before being washed with 6x with PBS–Tween PBS/T. Secondary antibody rabbit anti-human whole IgG conjugated to alkaline phosphatase (Sigma, USA) was added at a dilution of 1:3000 in casein–PBS solution and incubated for 1.5 hours at room temperature. After a final wash, plates were developed by adding 4-nitrophenyl phosphate substrate in diethanolamine buffer (Pierce, Loughborough, UK), and optical density OD was read at 405 nm using an ELx800 microplate reader (Cole Parmer, London, UK). A reference standard comprising of pooled cross-reactive serum and naïve serum on each plate served as positive and negative controls, respectively.

The positive reference standard was used on each plate to produce a standard curve. The standard was made from cross-reactive serum against each HA1 protein. It was added in duplicate at an initial dilution of 1:100 in casein–PBS solution and diluted twofold 10 times, starting with an arbitrary value of antibody units determined using the NIH standard calculator[38]. Three blank wells containing casein–PBS solution only and further three blank wells containing naïve human sera or plasma were used as negative controls. The mean of the OD values of the naïve sera was then subtracted from all OD values on each plate before triplicates were fitted to a four-parameter standard curve using the positive reference standard[38]. At least two technical replicates were performed to ensure reproducibility.

**Pseudotyped influenza virus production.** Pseudotyped lentiviruses displaying influenza HAs were produced by transfection of HEK 293 T/17 cells (ECCAC, Public Health England, UK) with 1.0 µg of gag/pol construct, p8.91, 1.5 µg of a luciferase reporter carrying construct, pCSFLW, 250 µg of TMPRSS4-expressing construct and 1.0 µg of HA glycoprotein-expressing construct[12]. Transfections were performed in 10 ml of media DMEM 10% FCS, 1% penicillin–streptomycin, 20% L-glutamate and left for 8 hours. One unit of endogenous NA (Sigma, USA) was added to 10 ml of new media to induce virus budding. Media was removed 48 hours post the induction of budding and filtered with a 0.45-µm syringe. The pseudotyped influenza viruses were stored at −80 °C.

The strain accession numbers used for production of the pseudotyped lentiviruses are A/California/04/2009 (AEE69009), A/Solomon Islands/3/2006 (ABU99109), A/USSR/90/1977 (AAA43240), A/Denver/1957 (ABD15258), A/Iowa/1943 (ABO38373), A/PR/8/1934 (CCH23213), A/WSN/1933 (ACF54598) and A/South Carolina/1/1918 (AAC57065).

The p8.91, pCSFLW and TMPRSS4-expressing construct were gifts from Dr Nigel Temperton, whilst the HA-expressing plasmids were either gifts from Dr. Temperton or produced through the cloning of GeneArt Strings (Thermo Fischer Scientific, USA) into the pI.18 expression vector (also a gift from Dr. Temperton). All plasmids are available on request.

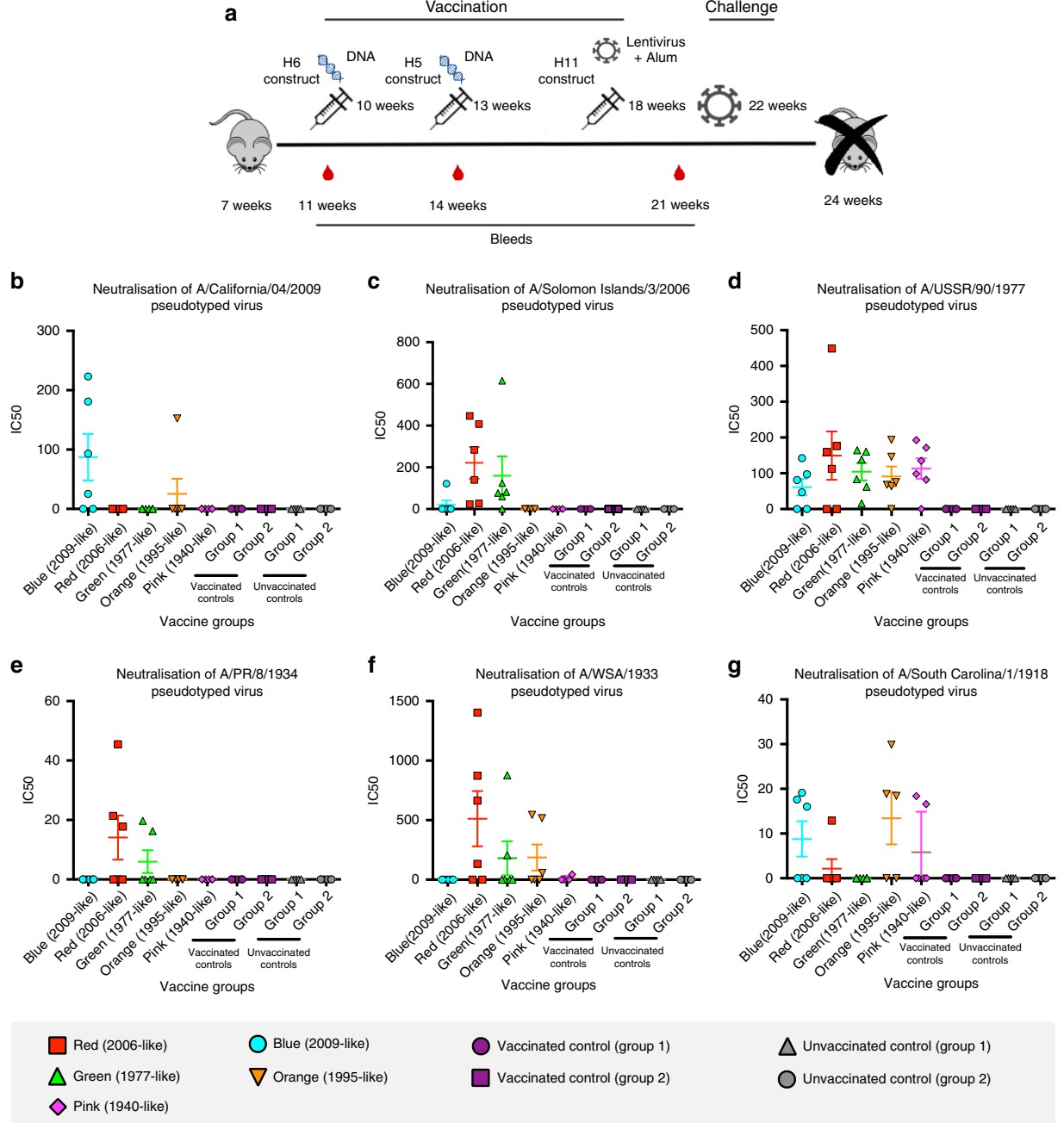

**Fig. 4** Sequential vaccination using chimeric HA constructs. **a** Five groups of mice were sequentially vaccinated with 2009-like (blue), 2006-like (red), 1995-like (orange), 1977-like (green) and 1940-like (pink) epitope sequences substituted into H6, H5 and H11 HAs (see Supplementary Table 1). Two further control groups were sequentially vaccinated with H6, H5 and H11 constructs without any sequence substituted into the HAs (vaccinated controls). Further two groups were mock vaccinated (unvaccinated controls). **b–g** Pseudotype microneutralisation assays using 0.5 µl of sera from the bleed at 21 weeks. Error bars are mean ± s.e.m. $n = 6$ for experimental groups and control groups. The values provided are an average of two replicates

**Pseudotyped influenza virus titration**. Serial dilutions were made of pseudotyped influenza virus preparations in Corning Costar plates 96-well plates (Promega, USA). A total of $10^4$ HEK 293 T/17 cells were added to each well and incubated for 3 days at 37 °C. The cells were then lysed with BrightGlo reagent (Promega, USA), and the relative light units of the cell lysate were determined using a Varioscan luminometer microplate reader (Thermo Fisher Scientific, USA).

**Pseudotype microneutralisation assay**. Neutralising antibodies were quantified using a pseudotype microneutralisation assay. Serially diluted sera was added to Corning Costar plates 96-well plates (Promega, USA) before being incubated with $10^6$ RLU-pseudotyped influenza virus for 1 hour at 37 °C. Each dilution was made in duplicate. A total of 2 µl of sera were used per replicate in Fig. 1a. For

comparison of WT and SDM-pseudotyped influenza viruses, 10 µl of sera was used per replicate in Fig. 1b–f. A total of 1 ul of sera per replicate was used in the vaccination experiment shown in Fig. 4 due to the small amounts of blood collected from the mice. HEK 293 T/17 cells $2.0*10^5$ cells ml$^{-1}$ were subsequently added to each well and incubated for 3 days at 37 °C. The cells were lysed with BrightGlo reagent (Promega), and the relative light units of the cell lysate were determined using a Varioscan luminometer microplate reader (Thermo Fisher Scientific, USA). The reduction of infectivity was determined by comparing the RLU in the presence and absence of antibodies and expressed as percentage neutralisation. The 50%-inhibitory dose, IC50, was defined as the sample concentration at which RLU were reduced 50% compared with virus control wells after subtraction of background RLU in cell-only control wells. At least two technical replicates were performed for each biological sample to ensure replicability.

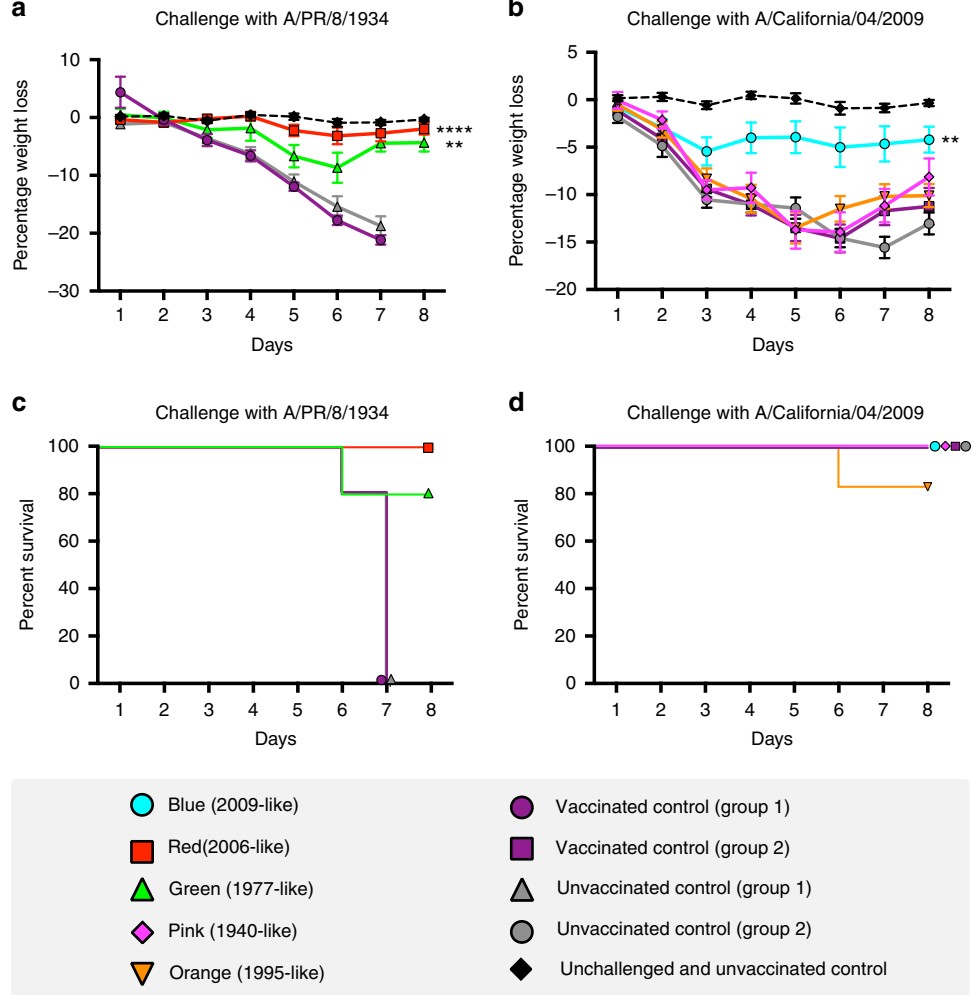

**Fig. 5** Influenza challenge of vaccinated mice. Mice were challenged with either A/PR/8/1934 (**a**, **c**) or A/California/4/2009 (**b**, **d**). **a**, **b** The graphs denote daily weight loss of the mice during the challenge. Mice of the same age, which were not vaccinated or challenged, are shown for reference and denoted unchallenged and unvaccinated. **c**, **d** Survival curves denoting the number of mice in each group. Mice were euthanised at 20% weight loss. Area under the curve was calculated for the mouse weight loss data and analysed in a single-factor ANOVA. Between-group comparisons were then performed using Tukey's post hoc method for pairwise comparison correction to provide corrected $p$-values. ****$p$-value < 0.0001 and **$p$-value < 0.010. Error bars are mean ± s.e.m. $n = 6$ for experimental groups and control groups

The C179 antibody used as a control in Fig. 1 b–f was obtained from Takara Bioscience (Product code: SD3145)

**Structural bioinformatic analysis.** Amino acids present on the surface of various H1 HAs were determined by calculating the accessibility of amino acids on the surface of the crystal structures of A/Brevig Mission/1/1918 (1RUZ[26] [https://www.rcsb.org/structure/1ruz]), A/Puerto Rico/8/1934 (1RU7[26] [https://www.rcsb.org/structure/1ru7]), A/California/4/2009 (3LZG[28] [https://www.rcsb.org/structure/3lzg]) and A/Washington/5/2011 (4LXV[29] [https://www.rcsb.org/structure/4LXV]) HAs using Swiss-Pdb viewer. Areas of 600, 800 and 1000 Å² were mapped onto the surface of the crystal structures by determining the distances between the α carbon of a given amino acid and all others within a structure. Those residues whose α carbon sequences were within the specified area were recorded and used to produce disrupted peptide sequences for a given binding site. ABS variability was calculated as the mean pairwise hamming distance between the consensus sequences collected between 1918 and 2016. The sequences were aligned using MUSCLE before being manually curated using AliView.

**Vaccination of mice.** Animal studies were approved by the University of Oxford Ethical Review Committee and were performed in strict accordance with the terms of a licence granted by the UK Home Office under the Animal (Scientific Procedures) Act 1986 (licence number: 30/2889).

Female BALB/c mice, $n = 6$, (bought from Envigo, UK) were sequentially vaccinated with the OREO sequences substituted into H6, H5 and H11 HA backbones in a prime–boost–boost regime at intervals of 3–4 weeks. As a backbone control, two groups were vaccinated with native H6, H5 and H11 constructs.

Further two groups were mock vaccinated and used as an unvaccinated control. The prime and first boost were administered as a 100-μg intramuscular injection of DNA into the musculus tibialis, whilst the final vaccination was administered as an intramuscular injection into the musculus tibialis of eight HI units of lentivirus pseudotype displaying the chimeric H11 HA in Alum adjuvant Alhydrogel (Invitrogen, USA) at a 1:1 volume ratio.

Individuals carrying out the mouse vaccination experiments were blinded regarding the vaccines being administered to the mice; vaccines and cages were numbered and administered as outlined above. The individuals carrying out the experiments were only notified of the vaccine identities after completion of the experiment. No exclusion criteria were applied to the mice and no randomisation was applied to the experiments as inbred mice were used.

**Haemagglutinin inhibition assay.** Pseudotyped lentivirus displaying influenza HA was diluted twofold down a 96-well plate and mixed with 50 μl of 4% chicken red blood cells. After an hour, the coagulation of red blood cells was assessed visually to determine the point at which coagulation could no longer be observed.

**Influenza challenge.** At 5 weeks post final boost, mice were challenged intranasally with either A/PR/8/1934 at $1.0 \times 10^3$ PFU per mouse or A/California/04/2009 at $1.5 \times 10^5$ PFU per mouse. Mice were weighed daily. At 3 days post-challenge, food in proportion to the number of mice in each cage was placed on the floor of the cage. Mice were euthanised at their pre-determined humane endpoint of 20% weight loss or if they showed no sign of recovery at 7 days post-challenge. Group sizes were determined using a power calculation based on data from the previous studies[38].

**Statistical analysis**. Student's *t* tests were performed to determine all *p*-values shown in Fig. 1.

Area under the curve was calculated for the mouse weight loss data (Fig. 5a and c, main text) and analysed in a single-factor ANOVA. Between-group comparisons were then performed using Tukey's post hoc method for pairwise comparison correction to provide the corrected p-values.

Fisher's exact test was used to determine survival differences in the experimental groups after 7 days (Fig. 5b and d). All *p*-values were adjusted to multiple comparisons using the Bonferroni–Holm correction.

**Phylogenetic analysis**. RAxML version 8.2.11 was used to build a maximum likelihood tree based on the strain HA amino acid sequences, using a gamma distributed site heterogeneity model and the amino acid FLU substitution model. Tip-to-root distance was regressed against sequence dates, using a best fitting root, in Tempest V1.5.1. This yielded an $R$-squared of 0.886 and 0.834 for the ≤ 2008 and ≥ 2009 data, respectively, indicating a good fit between the genetic distance and the time of sampling. The colour of branches was determined by the identification of amino acids at positions 147 and 158, which are the variable amino acids at the centre of the amino acid binding site. Blue OREO was defined as position 147 as lysine and position 158 as no lysine; orange OREO as position 147 as lysine and position 158 as lysine; green OREO was defined as position 147 as arginine; red OREO was defined as position 147 as a deletion; pink OREO was defined as position 147 as isoleucine.

Accession numbers can be found in the Supplementary Tables 1 and 2.

**Code availability**. The code will be made available from the corresponding authors to anybody on request.

## Data availability
The datasets generated or analysed during this study are available from the corresponding authors on request.

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

## Acknowledgements
We would like to thank Dr John S, Tregoning Imperial College for kindly providing us with the viruses for the influenza challenge. We thank the parents/guardians who gave written informed consent for use of these sera samples analysed in this study, with ethical approval by a local research ethics committee, South Central–Hampshire B Research Ethics Committee (ref: 16/SC/0141). Funding for the study was provided by a Royal Society Translation Award, a MRC confidence-in-concept grant and an ERC Advanced grant (DIVERSITY). O.G.P. and C.P.T, were supported by the European Union's Seventh Framework Programme (FP7/2007–2013)/European Research Council (614725-PATH-PHYLODYN). S.G, J.L and C.P.T were supported by the European Union's Seventh

Framework Programme (FP7/20–2013)/European Research Council)/European Research Council (268904-DIVERSITY). U.O. was supported through an EMBO fellowship. The Wellcome Centre for Human Genetics is supported by grant 203141/Z/16/Z. T.A.B. is supported by MRC grant MR/L009528/1.

## Author contributions

C.P.T undertook the structural bioinformatics analysis, in vitro experiments and mouse vaccination studies. J.L assisted with the structural bioinformatics analysis. A.A.W assisted with the mouse vaccination experiments. U.O assisted with the structural bioinformatics analysis, data analysis and phylogenetic analysis. M.E assisted with the in vitro assays. D.S.P assisted with the structural bioinformatics analysis. K.K assisted with the in vitro assays. G.W.C assisted with the in vitro assays. D.O'C. assisted with the experimental design and contributed reagents. T.A.B assisted with the structural bioinformatics analysis. O.G.P. assisted with the phylogenetic analysis. A.J.P assisted with the experimental design and contributed reagents. N.J.T assisted with the experimental design and contributed reagents. T.L assisted with experimental design and data analysis. S.C.G assisted with experimental design, and S.G assisted with experimental design and data analysis.

## Additional information

**Competing interests:** C.P.T and S.G are named as inventors on a patent application for an influenza vaccine targeting the epitope conformations outlined in this paper (PCT/GB2017/052510). It should be noted in the interests of full disclosure that S.C.G is a co-founder of Vaccitech, a spin-out company from the University of Oxford, which is developing an MVA-NP + M1 influenza vaccine. However, the MVA-NP + M1 vaccine as well as Vaccitech are entirely separate from and in no way are connected to the work undertaken in this paper as well as the vaccine proposed in the paper. All other authors report no potential conflicts.

