## [Peer Review File · Nature Communications]

Reviewers' comments:

Reviewer #1 (Remarks to the Author):

The manuscript entitled "A potent neutralizing epitope of limited variability in the head domain of haemagglutinin as a novel influenza vaccine target" by Thompson et al. describes a location on the H1 HA head that has evolved over time most likely to evade immune pressure. This site and region of HA has been known to researchers for several years now as a critical site for receptor binding site and antibody binding will neutralize or reduce viral infection. Alterations in this site result in antibody mediated escape. The authors used various epitopes in this region from historical H1 strains in a prime-boost-boost fashion using an irrelevant HA molecule to show that this site elicits breadth of HAI activity and that people, particularly children, have antibodies that bind this region.

None of this is surprising. using a heterologous prime boost strategy with full HA or HA1 proteins can achieve the same effect. The authors have shown that this region partially recapitulates this phenomenon is not stunning and does not mean that other regions or epitopes could achieve the same effect. The authors also bias the results by selecting strains in the HAI assay that are mostly "pandemic-like" that are antigenically similar, when in reality, they are all similar HA proteins. A more diverse panel of seasonal and pandemic like HA proteins and viruses would bolster their cause. Overall, this result is interesting, but only an incremental increase in our understanding of H1 HA neutralization and vaccine design.

Reviewer #2 (Remarks to the Author):

Thompson et al. present a fascinating study of the identification and characterisation of a potent neutralizing epitope with limited variability on the head of the HA protein of seasonal influenza H1 viruses, first identifying that sera collected from children in 2006/07 cross-reacted with historical strains and that this pattern of recognition is consistent with recycling of epitopes, and that vaccinating against this epitope provides a cyclical cross-reactivity to viruses collected across a century. A major conclusion from this study is that vaccinating against these conserved and immunogenic epitopes would allow to avoid yearly reformulation. The experimental work is mostly clearly described, and I agree with most of the conclusions of the manuscript, however there were a few points where the authors gloss over some obvious questions that arise or remain silent on the issue.

1. "All individuals possessed neutralising antibodies to the A/Solomon Islands/3/2006 strain and 98% of individuals possessed neutralising antibodies to A/New Caledonia/20/1999, including those children born in 2000 who are unlikely to have been exposed to the strain."

The A/New Caledonia/20/1999 strain was recommended as a seasonal influenza vaccine candidate in both hemispheres during 2000-2002/3 suggesting these were the dominant H1 strains circulating during this period. This contradicts with the statement, particularly, "including those children born in 2000 who are unlikely to have been exposed to the strain." Suggest removing or modifying.

2. "the existence of a lysine at position 147 may contribute to the overall lack of neutralisation of A/California/4/2009 and A/South Carolina/1/1918."

It is surprising that a Lysine at this position has a more drastic effect than a deletion at this position. Could this be due to the greater variation of Cal/4/09 and SC/1/18 across the HA (and adjacent to the site) than other H1N1 strains that were studied?

3. It is well established that H1N1/1977 viruses were similar to those that circulated during 1957, however the phylogenetic placement in Figure 4D does not reflect this – it rather suggests that they were derived from an earlier ancestor?

4. The conclusion of in-silico analysis that "there are numerous sites of limited variability in the head domain of the H1- HA" is not clear. I am not sure if I am missing something, but the study highlights few. Probably better to just state a number.

5. Essential experimental details that are not clear.

a. Lentivirus production. Due to the availability of a number of genetic variants of historical strains, especially PR8, it would be important to provide strain information (e.g. GB accession number) of strains used for pseudotyped lentivirus production.

b. It would be useful to make clear earlier in the ms that A/California/09 was a pandemic strain with independent origins.

6. It is not clear how these results would apply to the better studied H3N2 viruses as suggested in the conclusion of this study, particularly since the characteristic narrow genetic and antigenic diversity is exhibited predominantly by H3N2 and much less by H1N1.

Reviewer #3 (Remarks to the Author):

Summary

The authors employ a panel of human serum samples collected from individuals aged between 6 and 12 years in 2006/2007 in a pseudo-virus neutralization assay in order to measure the extent of cross-neutralizing activity present in the sera that is specific to a panel of influenza A virus strains isolated in the years spanning 1918 to 2009. Using this approach, coupled to an in silico analysis of the influenza HA protein structure, the authors identify an epitope in the globular head domain of the HA protein (coined "OREO"), which has displayed limited variability in terms of its amino acid composition, in strains isolated from 1918 to 2009. The epitope is responsible for a large proportion of the cross-neutralizing activity present in the serum samples tested. According to the authors' analysis, there are 5 major variants of the epitope that have arisen periodically in naturally circulating human strains of subtype H1 influenza A virus during the 20th and 21st century. The epitope encompasses, and includes, amino acids 147, 158 and 159 of the HA1. Presence or absence of the amino acid residue at position 147 is a strong determinant of sensitivity to, or escape from, any particular serum sample.

To assess the potential broadly neutralizing capacity of a vaccine focused on the OREO epitope, the following approach was taken: recombinant techniques were used to generate five distinct variants of the OREO epitope in each of three antigenically distinct HA protein backgrounds (H5 H6 H11). Groups of mice were vaccinated prime-boost-boost with each variant of the epitope, such that the dominant response to the immunization should be directed towards the relevant OREO epitope present in the vaccine. The authors show protection from weight loss and death in groups of OREO vaccinated mice that were challenged with distinct heterologous strains of influenza A that were isolated many years apart.

Main significance

The main significance of the paper lies in the successful efforts to stimulate the immune response in mice to a degree that allows protection across heterologous strains within the H1 subtype by focusing on a specific epitope in the HA globular head.

Drawbacks

The authors do not demonstrate that the vaccine confers immunity that is more recalcitrant to inducing escape variants of influenza A than conventional vaccine. This limits to some extent the impact of the study.

The authors should describe the extent of variability that has arisen in influenza A isolates spanning the 20th and 21st century in "conventional" antigenic sites, i.e those described by Caton et al., and compare the variability at these sites with that of the "OREO" site for comparison. i.e. what is the average variability at an antigenic site, and how much less variable is the OREO site to the other sites described?

The authors should describe the amino acid composition of the OREO epitope in the avian HA proteins used as control vaccines. Is this epitope conserved in the HA of all the avian strains used in the vaccines, and is it distinct to the human haplotypes described by the authors?

Response to reviewers' comments

We thank all the reviewers for their hard work in reviewing our manuscript and their comments. In general we found reviewers 2 and 3 to have provided very constructive and insightful comments. However, we reluctantly have to take substantial issue with reviewer 1's comments who unfortunately has not understood our methodology or approach.

Nevertheless, we have tried to address all the reviewers' comments as best as we can below. The reviewers' remarks are provided in regular font and our responses are provided in italics. Changes to the manuscript are highlighted in red.

Reviewers' comments:

Reviewer #1 (Remarks to the Author):

The manuscript entitled "A potent neutralizing epitope of limited variability in the head domain of haemagglutinin as a novel influenza vaccine target" by Thompson et al. describes a location on the H1 HA head that has evolved over time most likely to evade immune pressure. This site and region of HA has been known to researchers for several years now as a critical site for receptor binding site and antibody binding will neutralize or reduce viral infection.

This is completely untrue – we show, for the first time that an influenza epitope reappears and disappears in a cyclical fashion as influenza evolves. These findings fundamentally change our understanding of the antigenic evolution of influenza, which is currently believed to occur in a linear and not cyclical manner (eg. Koel et al. 2013). Reviewer 2 called this 'fascinating', which is typically the response we get at conferences/to our pre-print.

Alternations in this site result in antibody mediated escape. The authors used various epitopes in this region from historical H1 strains in a prime-boost-boost fashion using an irrelevant HA molecule to show that this site elicits breadth of HAI activity and that people, particularly children, have antibodies that bind this region.

*In this context, the HAs were simply used to hold the epitope in the correct conformation (a scaffold), so that the immune response can be focused onto the site – **this is a widely accepted vaccine strategy used previously** (eg. Krammer et al. 2013). Reviewer 1 has misunderstood the concept in its entirety. All the HA scaffolds need to do is hold the epitope in the correct conformation and vary between prime and boosts.*

*Furthermore, it is also important to highlight **that we have not used HAI assays as the reviewer states.** We used neutralisation assays and ELISAs.*

None of this is surprising. using a heterologous prime boost strategy with full HA or HA1 proteins can achieve the same effect. The authors have shown that this region partially recapitulates this phenomenon is not stunning and does not mean that other regions or epitopes could achieve the same effect.

*We would refer the reviewer to our control where we perform a prime-boost-boost with just the H5, H6 and H11 HAs, resulting in no immunity. **This is precisely why the control was included (Figures 4 & 5).***

The authors also bias the results by selecting strains in the HAI assay that are mostly "pandemic-like" that are antigenically similar, when in reality, they are all similar HA proteins. A more diverse panel of seasonal and pandemic like HA proteins and viruses would

bolster their cause. Overall, this result is interesting, but only an incremental increase in our understanding of H1 HA neutralization and vaccine design.

The strains we used in the paper circulated in 2009, 2006, 1999, 1977, 1957, 1943, 1934, 1933 and 1918 – spanning 80 years of H1N1 circulation. Reviewer 2 states that they have been “collected across a century”, whilst reviewer 3 describes these as “distinct heterologous strains”. Of these nine strains only two were “pandemic” strains: those from 2009 and 1918, which are antigenically similar. The other strains are not antigenically similar to the pandemic strains as can be seen in Figure 1. It is preposterous to say so.

In addition to this, reviewer 1 again states that we use HAI assays! We used neutralisation assays and viral challenge to determine that our findings were correct. In our opinion, this clearly demonstrates that the reviewer has fundamentally misunderstood our manuscript as they have not even correctly ascertained basic but important details such as the types of assays used.

Reviewer 2 (Remarks to the Author):

Thompson et al. present a fascinating study of the identification and characterisation of a potent neutralizing epitope with limited variability on the head of the HA protein of seasonal influenza H1 viruses, first identifying that sera collected from children in 2006/07 cross-reacted with historical strains and that this pattern of recognition is consistent with recycling of epitopes, and that vaccinating against this epitope provides a cyclical cross-reactivity to viruses collected across a century.

We thank the reviewer for these very supportive comments stating that the paper “presents a fascinating study”.

A major conclusion from this study is that vaccinating against these conserved and immunogenic epitopes would allow to avoid yearly reformulation. The experimental work is mostly clearly described, and I agree with most of the conclusions of the manuscript, however there were a few points where the authors gloss over some obvious questions that arise or remain silent on the issue.

We thank the reviewer for saying that they “agree with most of the conclusions of the manuscript”.

1. “All individuals possessed neutralising antibodies to the A/Solomon Islands/3/2006 strain and 98% of individuals possessed neutralising antibodies to A/New Caledonia/20/1999, including those children born in 2000 who are unlikely to have been exposed to the strain.”

“The A/New Caledonia/20/1999 strain was recommended as a seasonal influenza vaccine candidate in both hemispheres during 2000-2002/3 suggesting these were the dominant H1 strains circulating during this period. This contradicts with the statement, particularly, “including those children born in 2000 who are unlikely to have been exposed to the strain. Suggest removing or modifying.”

We have now removed this statement from the manuscript to avoid confusion.

2. “the existence of a lysine at position 147 may contribute to the overall lack of neutralisation of A/California/4/2009 and A/South Carolina/1/1918.”

“It is surprising that a Lysine at this position has a more drastic effect than a deletion at this position. Could this be due to the greater variation of Cal/4/09 and SC/1/18 across the HA (and adjacent to the site) that other H1N1 strains that were studied?”

The serological experiment using sera from children taken in 2006/7 shows that the addition of a lysine at position 147 reduces or removes neutralisation of the A/Solomon Island/3/2006, A/Denver/1957, A/Iowa/1943, A/PR/8/1934, A/WSN/33 pseudotyped lentiviruses. As the vaccination experiment mirrors these results, we can say that the presence of a lysine at position 147 (as shown in Figures 1 & 3) is one factor preventing antibodies raised against, for example, the 2006 conformation of OREO binding to the 2009 conformation.

However, as the reviewer correctly points out, in the serology experiment (Figure 1) we cannot rule out completely that the removal of the lysine residues (through a deletion) would enable neutralisation of the A/California/04/2009 and A/South Carolina/1/1918 lentiviruses by the 2006/7 cohort sera. The lack of neutralisation is also likely to due to the variation of A/California/04/09 and A/South Carolina/1/18 in other parts of the HA.

We thank the reviewer for their suggestion and we have modified the manuscript (line 145) to reflect this.

“These results imply that at least part of the cross-reactive neutralising immune response within this cohort is mediated through the recognition of an epitope that contains a deletion at position 147. Moreover, the existence of a lysine at position 147 may contribute to the overall lack of neutralisation of A/California/4/2009 and A/South Carolina/1/1918, **in addition to other variation across the HA.**”

3. It is well established that H1N1/1977 viruses were similar to those that circulated during 1957, however the phylogenetic placement in Figure 4D does not reflect this – it rather suggests that they were derived from an earlier ancestor?

We thank the reviewer for raising this important point. H1N1 influenza was re-introduced in 1977 by the artificial released of a H1N1 strain supposedly from a Russian or Chinese research laboratory that last circulated around 1949/50 (Kendal et al 1978, Webster et al, 1979). Consequently, whilst as the tree does appear odd, as reviewer 2 understandably points out, it is correct. We have added a short referral to this in the paper to avoid the confusion of readers. Please see lines 177/8.

“**Please note the re-introduction of H1N1 influenza in 1977 involved a strain which previously circulated in 1949/50^{3,4}**”

4. The conclusion of in-silico analysis that “there are numerous sites of limited variability in the head domain of the H1- HA” is not clear. I am not sure if I am missing something, but the study highlights few. Probably better to just state a number.

We would state that the local minima of the bar plot in Figure 2A would potentially be good candidates as sites of limited variability in the head of the H1 HA.

As the reviewer notes, we mentioned two sites in the paper. We focus our analysis on the site we have coined ‘OREO’ as its capacity to cycle appears very clearly due to the disappearance and reappearance of a deletion as well as its potential as a vaccine target. In the discussion, we refer to data from several papers, which when looked at in the context of our model, suggest that another site identified by our analysis, including position 180, could be of limited variability. We believe that there are potentially 2 or 3 other sites, but we have not experimentally verified that they are of limited variability yet.

We thank the reviewer for this comment and have tried to clarify this in the paper. Please see lines 183-185.

“This analysis demonstrates that there are numerous **potential** sites of limited variability in the head domain of the H1 HA **represented by the local minima in Figure 2A**, in addition to a range of highly variable sites; the antigenic trajectory of the latter has been tracked in detail by several previous studies^{1,5}”

5. Essential experimental details that are not clear.

a. Lentivirus production. Due to the availability of a number of genetic variants of historical strains, especially PR8, it would be important to provide stain information (e.g. GB accession number) of strains used for pseudotyped lentivirus production.

b. It would be useful to make clear earlier in the ms that A/California/09 was a pandemic strain with independent origins.

We thank the reviewer for these very useful suggestions and have modified the text of the manuscript.

To address the details regarding lentivirus production we added the access numbers of the strains at lines 382 to 385.

“The strain accession numbers used are for production of the pseudotyped lentiviruses are A/California/04/2009 (AEE69009), A/Solomon Islands/3/2006 (ABU99109), A/USSR/90/1977 (AAA43240), A/Denver/1957 (ABD15258), A/Iowa/1943 (ABO38373), A/PR/8/1934 (CCH23213), A/WSN/1933 (ACF54598), A/South Carolina/1/1918 (AAC57065).”

To address point b we added the below at lines 89 and 90.

“We tested the prediction that HA epitopes of limited variability exist by performing microneutralisation assays using pseudotyped lentiviruses displaying the H1 HA proteins from a panel of historical influenza isolates (hereafter described as pMN assays^{6,7}); with sera obtained in 2006/2007 in the UK from 88 children born between March 1994 and May 2000 (Fig 1A). **This panel consisted of pseudotyped lentiviruses displaying HAs from seasonal influenza strains as well as two pandemic strains: A/California/04/2009 and A/South Carolina/1/1918.**”

6. It is not clear how these results would apply to the better studied H3N2 viruses as suggested in the conclusion of this study, particularly since the characteristic narrow genetic and antigenic diversity is exhibited predominantly by H3N2 and much less by H1N1.

We thank the reviewer for their comment and have modified the text to address their query. Please see lines 339 to 341 below.

“Indeed, Zinder et al 2013 have shown that the phylodynamics of H3N2 influenza is easily reproduced using the ‘antigenic thrift’ framework.”

Reviewer #3 (Remarks to the Author):

Summary

The authors employ a panel of human serum samples collected from individuals aged between 6 and 12 years in 2006/2007 in a pseudo-virus neutralization assay in order to measure the extent of cross-neutralizing activity present in the sera that is specific to a panel of influenza A virus strains isolated in the years spanning 1918 to 2009. Using this approach,

coupled to an in silico analysis of the influenza HA protein structure, the authors identify an epitope in the globular head domain of the HA protein (coined “OREO”), which has displayed limited variability in terms of its amino acid composition, in strains isolated from 1918 to 2009. The epitope is responsible for a large proportion of the cross-neutralizing activity present in the serum samples tested. According to the authors’ analysis, there are 5 major variants of the epitope that have arisen periodically in naturally circulating human strains of subtype H1 influenza A virus during the 20th and 21st century. The epitope encompasses, and includes, amino acids 147, 158 and 159 of the HA1. Presence of absence of the amino acid residue at position 147 is a strong determinant of sensitivity to, or escape from, any particular serum sample.

To assess the potential broadly neutralizing capacity of a vaccine focused on the OREO epitope, the following approach was taken: recombinant techniques were used to generate five distinct variants of the OREO epitope in each of three antigenically distinct HA protein backgrounds (H5 H6 H11). Groups of mice were vaccinated prime-boost-boost with each variant of the epitope, such that the dominant response to the immunization should be directed towards the relevant OREO epitope present in the vaccine. The authors show protection from weight loss and death in groups of OREO vaccinated mice that were challenged with distinct heterologous strains of influenza A that were isolated many years apart.

Main significance

The main significance of the paper lies in the successful efforts to stimulate the immune response in mice to a degree that allows protection across heterologous strains within the H1 subtype by focusing on a specific epitope in the HA globular head.

Drawbacks

The authors do not demonstrate that the vaccine confers immunity that is more recalcitrant to inducing escape variants of influenza A than conventional vaccine. This limits to some extent the impact of the study.

We thank the reviewer for raising this point and we have tried to clarify this basis further in the text in lines 309-336. We have also added supplementary figure S5 to support the statements made in the text below.

“Currently available influenza vaccines are believed to target epitopes of very high variability on the haemagglutinin and neuraminidase surface glycoproteins. This requires them to be continuously updated, with the only alternative being seen as the artificial boosting of immunity to conserved epitopes of low immunogenicity. By identifying such epitopes, we have established an alternate method of producing improved influenza vaccines: targeting highly immunogenic epitopes of limited variability as opposed to targeting highly immunogenic epitopes of high variability or conserved epitopes of low immunogenicity. Through vaccination against the various conformations of the epitope of limited variability identified in this study, it is possible to induce immunity to all previous and future H1N1 strains.

The OREO site has been under selective pressure over a period of 80 years, providing us the opportunity to observe and document its variation: the site has historically cycled between four conformations when position 147 is present and one conformation where it has undergone a deletion. Once the deletion has occurred, instead of the epitope varying further, on two occasions the circulating seasonal H1N1 strain died out – once in 1957 and again in 2009. In 2009, the seasonal influenza strain was replaced through zoonotic spillover with a pandemic H1N1 strain displaying a previously seen conformation of OREO to which immunity in 2009 has not yet built up against^{8,9}. In 1957, the seasonal influenza strain was replaced with H2N2 influenza (Figure 2D)¹⁰. These observations suggest that the antibodies

generated against these five conformations cover all possible variations within the OREO epitope that are found in evolutionary 'fit' H1N1 influenza viruses.

This limited variation exhibited by the epitope is determined by the composition of the site. Position 147 is next to, and affects the polarity of, the receptor binding site, and therefore there is a limited repertoire of amino acids it can cycle between. Furthermore, the OREO site is composed of a smaller number of residues compared other sites containing position 147 (see Figure 2A), which intrinsically limits its overall variability (Figure S4 A&B).

All the conformations of the OREO epitope could be displayed in a single vaccine by itself, or in concert with other epitopes of limited variability to create a single 'universal' influenza vaccine. Alternatively, the epitopes could be displayed in individual vaccines and deployed when the circulating conformation 'changes' (currently the OREO epitope changes roughly every 10 years, see Figure S3). It is critical feature of both approaches described above, is that the OREO epitope does not 'drift' to the same level as the currently targeted highly variable epitopes. Hence, both of these approaches could provide longer lasting vaccines in comparison to the trivalent and quadrivalent vaccines. Moreover, the longer cycling period of OREO might help decrease and also avoid mistakes in the formation of the vaccines, which sometimes occur due to formulation decisions having to be made at least 6 months prior to the influenza season¹¹."

The authors should describe the extent of variability that has arisen in influenza A isolates spanning the 20th and 21st century in "conventional" antigenic sites, i.e those described by Caton et al., and compare the variability at these sites with that of the "OREO" site for comparison. i.e. what is the average variability at an antigenic site, and how much less variable is the OREO site to the other sites described?

We thank the reviewer for their very useful suggestion and have added this information to the supplementary data section of the paper. Please see Supplementary Figure S5.

In this figure we have included the Sa, Sb, Ca₁, Ca₂ and Cb antigenic sites as defined by Caton et al, 1982 for A/PR/8/1934 (Mount Sinai) (Figure S5A). Matsuzaki et al, 2014 redefined the antigenic sites for A/Narita/1/2009 and we have included those sites too (Figure S5B). These sites were defined by selecting for escape mutant using monoclonal antibodies and it is important to point out it is unlikely that all the escape mutants isolated would be 'evolutionarily fit'.

Matsuzaki et al, substantially expanded and modified the Sa, Sb and Ca₂ sites by analysing 566 escape mutants but did not find any MAb that bound to the Ca₁ or Cb antigenic sites (shown in Figure 1 in Matsuzaki et al). Consequently, we would expect that neither study comprehensively maps the residues contained within epitopes in the head domain of the HI HA. For example, the Ca₁ sites defined by Caton et al contains just four residues and is much too small to be an epitope. The Pa antigenic site defined by Matsuzaki et al contains just a single residue (position 147).

Analysis of natural variation within the sites indicates that some of these antigenic sites appear to have limited variability (partly due to the small number of residues contained within them). This is in line with the predictions of our model (Recker et al 2007). However, until we do a full structural bioinformatics analysis, we cannot comment on the true extent of variability of the actual ABS containing these antigenic sites.

It should also be noted that OREO consists of the entire Ca₂ and Pa antigenic sites as defined by Matsuzaki et al, whilst it contains part of the Ca₂ antigenic site as defined by Caton et al.

The authors should describe the amino acid composition of the OREO epitope in the avian HA proteins used as control vaccines. Is this epitope conserved in the HA of all the avian strains used in the vaccines, and is it distinct to the human haplotypes described by the authors?

We thank the reviewer for their comments. The compositions of the 'equivalent' regions of the avian HAs used as vaccine scaffolds (Figures 4 & 5) can be found in Supplementary Table S1.

1. Koel, B. F. *et al.* Substitutions Near the Receptor Binding Site Determine Major Antigenic Change During Influenza Virus Evolution. *Science* (80-.). **342**, 976–979 (2013).
2. Krammer, F., Pica, N., Hai, R., Margine, I. & Palese, P. Chimeric Hemagglutinin Influenza Virus Vaccine Constructs Elicit Broadly Protective Stalk-Specific Antibodies. *J. Virol.* **87**, 6542–6550 (2013).
3. Webster, R. G., Kendal, A. P. & Gerhard, W. Analysis of antigenic drift in recently isolated influenza A (H1N1) viruses using monoclonal antibody preparations. *Virology* **96**, 258–264 (1979).
4. Kendal, A. P., Noble, G. R., Skehel, J. J. & Dowdle, W. R. Antigenic similarity of influenza A(H1N1) viruses from epidemics in 1977-1978 to 'Scandinavian' strains isolated in epidemics of 1950-1951. *Virology* **89**, 632–636 (1978).
5. Caton, A. J., Brownlee, G. G., Yewdell, J. W. & Gerhard, W. The antigenic structure of the influenza virus A/PR/8/34 hemagglutinin (H1 subtype). *Cell* **31**, 417–427 (1982).
6. Matsuzaki, Y. *et al.* Epitope Mapping of the Hemagglutinin Molecule of A/(H1N1)pdm09 Influenza Virus by Using Monoclonal Antibody Escape Mutants. *J. Virol.* **88**, 12364–12373 (2014).
7. Recker, M., Pybus, O. G., Nee, S. & Gupta, S. The generation of influenza outbreaks by a network of host immune responses against a limited set of antigenic types. *Proc. Natl. Acad. Sci.* **104**, 7711–7716 (2007).

REVIEWERS' COMMENTS:

Reviewer #2 (Remarks to the Author):

The authors have adequately address each of my comments, and have made adequate changes to the ms. The ms reads well (expect for a few minor typo's) - I have no further comments.

Reviewer #3 (Remarks to the Author):

No further concerns